# Storable Cell-Laden Alginate Based Bioinks for 3D Biofabrication

**DOI:** 10.3390/bioengineering10010023

**Published:** 2022-12-23

**Authors:** Anastassia Kostenko, Che J. Connon, Stephen Swioklo

**Affiliations:** 1Atelerix Ltd, The Biosphere, Draymans Way, Newcastle Helix, Newcastle upon Tyne NE4 5BX, UK; 2International Centre for Life, Faculty of Medicine, Bioscience Institute, Newcastle University, Central Parkway, Newcastle upon Tyne NE1 3BZ, UK

**Keywords:** alginate, biofabrication, biological preservation, mesenchymal stem cell, 3D printing

## Abstract

Over the last decade, progress in three dimensional (3D) bioprinting has advanced considerably. The ability to fabricate complex 3D structures containing live cells for drug discovery and tissue engineering has huge potential. To realise successful clinical translation, biologistics need to be considered. Refinements in the storage and transportation process from sites of manufacture to the clinic will enhance the success of future clinical translation. One of the most important components for successful 3D printing is the ‘bioink’, the cell-laden biomaterial used to create the printed structure. Hydrogels are favoured bioinks used in extrusion-based bioprinting. Alginate, a natural biopolymer, has been widely used due to its biocompatibility, tunable properties, rapid gelation, low cost, and easy modification to direct cell behaviour. Alginate has previously demonstrated the ability to preserve cell viability and function during controlled room temperature (CRT) storage and shipment. The novelty of this research lies in the development of a simple and cost-effective hermetic system whereby alginate-encapsulated cells can be stored at CRT before being reformulated into an extrudable bioink for on-demand 3D bioprinting of cell-laden constructs. To our knowledge the use of the same biomaterial (alginate) for storage and on-demand 3D bio-printing of cells has not been previously investigated. A straightforward four-step process was used where crosslinked alginate containing human adipose-derived stem cells was stored at CRT before degelation and subsequent mixing with a second alginate. The printability of the resulting bioink, using an extrusion-based bioprinter, was found to be dependent upon the concentration of the second alginate, with 4 and 5% (*w*/*v*) being optimal. Following storage at 15 °C for one week, alginate-encapsulated human adipose-derived stem cells exhibited a high viable cell recovery of 88 ± 18%. Stored cells subsequently printed within 3D lattice constructs, exhibited excellent post-print viability and even distribution. This represents a simple, adaptable method by which room temperature storage and biofabrication can be integrated for on-demand bioprinting.

## 1. Introduction

The ability to fabricate complex, three-dimensional (3D) biological structures for tis-sue engineering applications has advanced in recent years due to the emergence of 3D bioprinting. 3D bioprinting is an emerging technology that holds huge promise for many bio-medical applications, from fabricating models for fundamental biological research and drug testing, to the generation of new tissues and organs for clinical use [1,2,3,4].Various bioprinting techniques, laser-assisted bioprinting [5], extrusion bioprinting [6] and inkjet bioprinting [7], have been designed to deposit or ‘print’ bioink, an extrudable biomaterial containing cells, in the distinct form of a desired tissue scaffold or construct. Successful bioprinting is dependent on the bioink, the material that contains cells that can be accurately positioned in a spatiotemporal manner. Engineering a suitable bioink is arguably one of the most difficult challenges in the bioprinting process as it requires the optimisation of several properties. It is important for example, that the bioink flow sufficiently as to ensure printability, yet not too viscous that its extrusion causes the cells within the bioink to shear [8]. The bioink must additionally be able to undergo chemical modifications (so as to regulate its microenvironment subsequent to being printed) and demonstrate good biocompatibility [9,10]. For these reasons, hydrogels derived from natural polymers have been a favoured choice of biomaterial for tissue engineering purposes. Hydrogels can be engineered to display similar structural properties to that of the extracellular matrix (ECM) of soft tissue due to their remarkable ability to absorb vast quantities of water relative to their dry weight [9,11]. By adjusting variables such as temperature, pH, crosslinking density and polymer composition, the mechanical properties of hydrogels can be modified to suit the required mechanical performance of the scaffold [12,13]. Of these hydrogels, alginate has been widely used for biofabrication due to its excellent biocompatibility, tunable proper-ties, rapid gelation, low cost, and easy incorporation with other matrices or bioactive peptides [8,14,15,16].

The need to maintain good post-printing cell viability presents a further factor to be considered with respect to bioink manufacture. Studies have shown that material flow rate, dispensing pressure, nozzle geometry and material concentration are among the factors which affect cell viability during the 3D bioprinting process [17]. It is therefore ideal for cells to be incorporated into the bioink just before bioprinting begins in order to maximise the number of viable cells post printing. A consequence of this is that harvesting of cells is often carried out just prior to their incorporation into the biomaterial, which can be a lengthy process especially when desired cell concentration is high. Thus, a storable bioink in which cells remain viable at room temperature for extended periods of time (days) would therefore be highly beneficial for the tissue 3D bioprinting process. In order to realise full clinical translatability, however, biologistics need to be considered. Similar to the cell therapy supply chain, the manufacture of cell therapy products (CTPs) for clinical application can face a number of hurdles. The complex supply chain can involve a number of sites of manufacture/processing and tight scheduling between these sites and the clinic [18,19]. The transport of living biologics around the supply chain is key, and inadequate scheduling can lead to high wastage and subsequent monetary cost.

Previous research conducted in our laboratory has shown that alginate can be used as a biomaterial in which cells can be stored under hypothermic conditions [20]. The optimal storage temperature for viable cell recovery after 72 h was found to be 15 °C, whereupon recovered cells were observed to retain a normal phenotype, metabolic activity and trilineage differentiation. In the present study, we exploited the dual functionality of alginate as a biomaterial for hypothermic cell-storage as well as its suitability for use as a bioink for on-demand 3D bioprinting of live cells. To demonstrate this, we have developed a four-step process where crosslinked alginate containing cells could be stored at controlled room temperature (CRT) conditions, before being reformulated to a printable alginate-based bioink. Different formulations were examined for their printability as well as their ability to support the printing of 3D constructs from bioinks containing cells that had been stored for 1 week at CRT. Whilst bioprinting requires a viscous, free-flowing alginate matrix for the fabrication of complex structures, the protection offered by alginate during storage requires the gel to be crosslinked with divalent calcium cations. It was therefore necessary to de-crosslink this gel through the addition of a chelator. In this form, however, the material was not suitable for extrusion-based printing due to its low viscosity. To overcome this, we added a reduced volume and concentration of the chelator, before the addition of a second, high viscosity, alginate (HV-alginate). The concentration of HV-alginate required was optimised and the effect that the presence of the chelator would have on the material properties of the reformulated alginate and its ability to re-gel was investigated. Taken together, we have established a method by which room temperature storage and biofabrication can be integrated for on-demand printing.

## 2. Materials and Methods

### 2.1. Bioink Preparation

Bioinks were prepared using a two-step process. BeadReady™ product, available from Atelerix Ltd. was used to store cells under hypothermic conditions. Subsequently, the cell-laden beads were dissolved in an equal volume of 50 mM trisodium citrate (Sigma-Aldrich) before combining with a second type of high viscosity sodium alginate (HV-alginate) (Acros Organics brand, ThermoFisher Scientific, Loughborough, UK) to increase ink viscosity following storage and dissolution with the aim to aid in bioprinting. HV-alginate was used at a specified concentration between 2–10% to reach final concentrations of 1–5%. These were used to compose a range of bioinks at various concentrations in order to evaluate their suitability for 3D bioprinting applications.

### 2.2. Evaluation of Viscosity, Rheology and Printability

For the examination of relative viscosity, 300 μL drops of each bioink sample (prepared as per 2.1) were placed in ascending order of HV-alginate concentration on the broad/upper edge of a glass slide. The glass slide was placed inside a sealed, humidified Petri dish and inclined at an angle of 110° for 15 min. Drop displacement was measured as the distance between the initial centre of each drop before inclination and its associated tip/apex after inclination using Image J (1.48v) software (Schneider et al., 2012).

Rheological assessment and viscometry measurements of each bioink sample (1–5% HV-alginate) (prepared as per 2.1) was conducted using the Kinexus pro+ rotational rheometer (Malvern instruments Ltd., Malvern, UK). Parallel plate geometry (CP1/60 60 mm diameter with a 1 mm gap) with controlled shear strain was used. All measurements were performed at 25 °C and rSpace software was used for data analysis. Rheological data including the shear elastic modulus G′, shear loss modulus G″, and loss tangent tan(δ) was collected from the oscillation frequency sweep, conducted over a frequency range 0.1–10 Hz. The individual test samples were dispensed on the bottom plate geometry and trimmed, once the upper geometry was in place. Each experiment was conducted six times and the mean and standard deviation (SD) were calculated and are presented. The dynamic viscosity of samples was evaluated by performing viscometry ramp measurements over a range of shear frequencies (0–587.9 y.(s^−1^)) at 25 °C and rSpace software was used for data analysis.

For the examination of printability, exemplary 3-dimensional lattice constructs were printed using bioinks made with 3, 3.5, 4, 4.5 and 5% (*w*/*v*) HV-alginate. Constructs were printed using a pneumatic-based extrusion INKREDIBLE 3D Bioprinter (CELLINK AB, Gothenburg, Sweden) at approximate air pressures of 65, 80, 85, 100 and 125 KPa, for 3, 3.5, 4, 4.5 and 5% (*w*/*v*) HV-alginate, respectively. Pressure was adjusted as necessary in order to ensure continuous flow of the filament. In each case, the bioink was extruded through a 0.25 mm nozzle/needle. Gross images of prints were captured before crosslinking with 102 mM calcium chloride. Following a brief wash with HBSS, phase contract images of the 3D-printed constructs were captured using a Leica DM IL LED microscope (Leica, Milton Keynes, UK). Phase contrast images were used to evaluate printability as a measure of average extruded thread width, measured using ImageJ (1.48v) software.

### 2.3. Cell Culture

Human adipose-derived stem cells (hASCs) (Invitrogen, Glasgow, UK) obtained commercially from the subcutaneous fat of 3 healthy donors (aged 45–63 years) were used for all experiments. Following recovery from cryostorage, cells were seeded at 800 cells/cm^2^ and maintained in growth medium in a humidified incubator at 37 °C, 5% CO_2_ with medium changes every 3–4 days until approximately 80% confluence was reached. Cells were harvested using TrypLE Express enzyme (ThermoFisher Scientific, Loughborough, UK). Cells for all experiments were used at passage 4.

### 2.4. Cell Encapsulation, Storage and 3D Bioprinting

Alginate beads (0.5 mL) containing 2 × 10^6^ hASCs were prepared as per standard BeadReady™ protocol supplied by Atelerix Ltd. (https://www.atelerix.co.uk/products/suspended-cells-beadready/ (accessed on 1 October 2022)). Briefly, a cell suspension of 8 × 10^6^ cells/mL in 0.25 mL growth medium was mixed with 0.25 mL of Component A before deposition and gelation with Component B provided in the BeadReady™ kit for 8 min. Following a 2 min was with culture medium, beads were either transferred for storage for 7 days at 15 °C in a cooled incubator (INCU-Line, VWR Collection, Leicestershire, UK) (stored) or placed in growth medium ready for printing (non-stored). Non-stored samples were dissolved in 0.5 mL 50 mM trisodium citrate and viable cell number was assessed (as described in 2.5) before mixing with 1 mL HV-sodium alginate (Acros brand) at concentrations of 8% and 10% (*w*/*v*). The cell-containing ink was subsequently loaded into bioprinting cartridges (CELLINK AB) and 5 lattice-constructs of each bioink were printed using air pressures of approximately of 85 KPa (4% HV-Alginate) and 125 KPa (5% HV-Alginate) through a 25G high precision conical bioprinting nozzle (CELLINK, AB). Constructs were subsequently crosslinked using 500 μL of 102 mM CaCl2 for 8 min, washed briefly in growth medium, and cultured for 14 days in 3 mL medium under normal cell culture conditions, with medium replenishment every 3–4 days. This process was repeated for stored samples. Cell encapsulation was carried out three times in total using cells from three different donors.

### 2.5. Assessment of Live Cells Prior to Printing

Viable cell number was assessed following gel dissolution by live-dead staining. Cells were stained with 1 µM Calcein-AM (eBioscience brand, ThermoFisher Scientific, Loughborough, UK) and 2 µM Ethidium Homodimer 1 (Sigma-Aldrich, Gillingham, UK) for 15 min at 37 °C. Viable cell number and percentage viability were assessed using a Countess II FL automated cell counter (Invitrogen brand, ThermoFisher Scientific).

### 2.6. Assessment of Post-Printing Cell Viability and Distribution

The printed constructs were assessed for cell viability and cell distribution on days 0, 1, 4, 7 and 14 subsequent to printing. Printed constructs were washed in HBSS before suspending in 0.5 mL HBSS containing 1 µM Calcein-AM (eBioscience brand, ThermoFisher Scientific, Loughborough, UK) and 2 µM Ethidium Homodimer 1 (Sigma-Aldrich, Gillingham, UK). Constructs were incubated for 30 min at 37 °C before washing twice with HBSS and capturing fluorescent images using a Leica DM IL LED microscope (Leica, UK). Cell viability was calculated from images using ImageJ (1.48v) software (Schneider et al., 2012). Gross images were also captured on each of these days to establish printed construct stability.

## 3. Results

### 3.1. Effect of Alginate Concentration on Bioink Viscosity

Schematic representation of the process of cell encapsulation, storage, dissolution and printing is summarized in Figure 1. After the de-gellation of the cell-laden storage alginate and mixing with a high viscosity alginate, the comparative viscosity of the resulting bioinks was assessed by their ability for drops to hold their form on an inverted glass slide (Figure 2a). It can be visually observed that HV-alginate drops with concentrations of 1, 1.5, and 2% *w*/*v* moved significantly further compared to the 3.5, 4, 4.5 and 5% *w*/*v* HV-alginate solutions. These results are supported by the distance measurements where alginates at final concentrations of less than 2% (*w*/*v*) (1, 1.5, 2%) demonstrated a significant (*p* ≤0.01, *p* = < 0.05, *p* = < 0.05, respectively) increase in the distance the drop travelled after 15 min of inversion (Figure 2b). Whilst not statistically significant, drops of 2.5% (*w*/*v*) alginate did not hold their drop form after inversion and had started to flow down the slide. As such, it was deemed that a final concentration of between 3 and 5% (*w*/*v*) was appropriate for printing, where the drop-shape was maintained.

Further assessment of viscosity of the samples has been performed using the Kinexus pro+ rotational rheometer (Malvern instruments Ltd., Malvern, UK) over a range of shear rates (0–587.9 y.(s^−1^)). Figure 3a illustrates the obtained dynamic viscosity measurements where the measured viscosity in Pas is presented on the y axis and the shear rate (y.(s^−1^)) is presented on the x axis with the percentage of alginate tested indicated by different colours on the graph and in the figure legend. The data represents means ± SD from 6 independent samples. Figure 3b displays the viscosity of tested alginates at the lowest shear rate (fourth row from the top) and the highest shear rate (fifth row from the top) for easier comparison. Since the rates of shear were different for every tested sample they are also displayed in the table. The lowest shear rate is displayed in the second row and the highest shear rate is displayed in the third row. First row corresponds to the concentrations of alginate assessed. It can be observed that the alginate samples of higher percentage require higher amounts of shear to diminish in viscosity. A clear trend is evident, viscosity decreases with increasing shear rate in all samples tested. Higher shear rates are required to decrease the viscosity of solutions with a higher concentration. As expected, solutions with a lower concentration of alginate exhibit more pronounced drops in viscosity compared to solutions of a higher concentration.

The viscometry results are further supported by rheological assessment of the samples whereby the storage and loss moduli of the materials were determined in frequency sweep tests across the frequency range of 0.1–10 Hz. The storage moduli (G′, in Pa) of all the alginates tested are presented in Figure 4 on the log y axis, while frequency of oscillation is plotted on the x axis. The loss moduli (G″ in Pa) are exhibited in Figure 4 on the same log y axis as the G′ values.

The storage modulus (also called the elastic modulus) is a measure of the elastic component of the viscoelastic sample, which corresponds to the solid-like behaviour of the sample. The loss modulus is an indication of the viscous component of the material and represents the liquid-like behaviour of the viscoelastic sample. Viscoelastic solids exhibit a higher storage modulus compared to the loss modulus (G′ > G″), because of the strong internal chemical bonds and physio-chemical interactions in the sample. Viscoelastic liquids display a greater loss modulus compared to the storage modulus (G″ > G′), because of the lack of robust bonds between the individual particles within the sample. The phase angle (tan, δ) is a factor that depicts the ratio of the elastic behaviour to viscous behaviour in a viscoelastic sample. The phase angle measurements can range between 0 and 90 degrees, whereby 90 degrees demonstrates an ideal liquid (water has a phase angle of 90 degrees) and 0 degrees and ideal solid. For ideally elastic behaviour there is no viscous component (G″ = 0). Ideally viscous behaviour corresponds to δ = 90°, this is because G′ = 0, meaning there is no elastic component in the sample. The phase angle can indicate the phase transition (sol-gel transition point) of the sample that happens during measurement.

Figure 4 indicates that storage modulus values of increasing concentrations of alginate solutions increase with increasing frequency in all the samples tested. The same material behaviour can be seen in relation to the loss moduli. For instance, 1% alginate solution has a storage modulus of 0.05617 Pa at 0.1 Hz, which reaches up to 7.362 Pa at Hz of 10. These values are lower than the loss modulus (G″) at the lowest shear frequency (G″ of 1% alginate at 0.1 Hz is 0.5327 Pa). The storage modulus does not exceed the loss modulus at any frequency, suggesting primarily liquid like behaviour of the material. A similar pattern was observed in 1.5% alginate where G″ exceeded G′ at all frequencies tested. The values of storage and loss moduli increase with increasing alginate concentrations. Storage and loss moduli of 2% alginate solution increase further compared to 1% and 1.5%. At the concentration of 2% G′ (125.8 Pa) exceeds G″ (124.3 Pa) at the frequency of 8.031 Hz, indicating primarily liquid like properties of the material up until that critical strain point, where a transition towards more solid like properties occurs. This trend is evident in all other alginate samples, with increasing concentration of the sample the frequency at which the storage modulus exceeds the loss modulus decreases. Overall, alginates with higher concentrations have higher loss and storage modulus values and lower phase angle values due to the greater viscosity of the resulting solutions. For a clearer comparison between the conditions the storage and loss modulus values as well as phase angle values of all alginate concentrations are presented in Table 1. Concentration of alginate is displayed in the top row with the obtained G′ values at 0.1 Hz in the first row, G′ values at 10 Hz in the second row, G″ values at the lowest frequency tested in the third row, G″ values at the highest frequency tested in the fourth row as well as obtained phase angle values at 0.1 Hz in the fifth row and at 10 Hz in the last row. The storage and loss moduli of all the tested solutions increase at higher frequencies and with increasing alginate concentration while the phase angle values decrease with increasing shear frequencies and with increasing alginate concentration. Additionally, for a clearer comparison of phase change points (where the G′ exceeds the G″) Appendix A displaying the loss and storage moduli of each alginate concentration tested in a separate graph has been included.

### 3.2. Examination of Bioink Printability

Following the selection of a suitable range of HV-alginate concentrations, bioink printability was examined. BeadReady™ beads were dissolved in an equal volume of 50 mM sodium citrate before mixing with an equal volume of HV-alginate at final concentrations of 3, 3.5, 4, 4.5 and 5% (*w*/*v*). 3D lattice constructs were subsequently printed using an extrusion-based bioprinter. At 3 and 3.5% (*w*/*v*) HV-alginate concentrations, the bioink was not viscous enough to form threads necessary to print the constructs. The printing fidelity was poor and the bioink was not able to hold its shape (Figure 5a), resulting in rapid spreading of the bioink, irregularity, and a significant increase in thread width (Figure 5(bi,bii)). Contrary to this, when final HV-alginate concentrations of greater than 4% (*w*/*v*) were used, printing fidelity was good (Figure 5a) with the highest HV-alginate concentration giving the best result. Upon microscopic inspection, 4 and 4.5% (*w*/*v*) HV-alginate resulted in some irregularity in the shape of the apertures in the lattice construct, whilst 5% (*w*/*v*) HV-alginate resulted in a highly regular print (Figure 5(bi)). Regardless of this, the thread width in printed constructs was not considerably different between the upper concentrations of between 4 and 5% (*w*/*v*) (Figure 5(bii)).

Whilst 5% (*w*/*v*) HV-alginate resulted in the best printing fidelity, high alginate concentrations can result in small pore sizes in the hydrogel when it is crosslinked. This could potentially restrict the mass transfer of waste and nutrients, and therefore compromise the viability of encapsulated cells. In addition to this, the higher force required to extrude the viscous bioink may result in increased levels of shear stress that could have a detrimental effect on post-print cell viability. We, therefore, selected 4 and 5% (*w*/*v*) concentrations to print 3D lattice constructs with bioinks containing encapsulated hASCs. We found that a concentration of between 4 and 5% (*w*/*v*) HV-alginate facilitated both rapid mixing and the ability to bioprint structures with high shape fidelity. These were then crosslinked for a second time forming stable hydrogel constructs.

### 3.3. 3D bioprinting of Storable Cell-Laden Bioinks

For the storage of hASCs, cells were encapsulated in BeadReady™ at a density of 4 × 10^6^ cells/mL and stored for 7 days at 15 °C. Following gel dissolution, a sample of cell suspension was collected for the enumeration of viable cells, and compared with encapsulated cells that had not been stored. Non-stored cells were cultured under standard cell culture conditions (in a humidified incubator with 5% CO_2_, at 37 °C). Viable cell number following storage and alginate dissolution was 1.89 ± 0.34 × 10^6^ cells, equating to a cell load 88 ± 18% of the non-stored equivalent (Figure 6(ai)). The percentage viability of cell populations was 93 ± 3% and 96 ± 3% for stored and non-stored samples, respectively (Figure 6(aii)). hASCs suspended in dissolved beads and sodium citrate were subsequently mixed with an equal volume of HV-alginate (as described in Figure 1) to result in final concentrations of 4% and 5% (*w*/*v*).

Immediately after printing, cell viability was high in all samples with percentage viabilities of between 94% and 97% (Figure 6b). Cells were evenly distributed throughout the printed constructs with no evidence of clumping (Figure 6c). 24 h after printing and return to normal culture conditions, cell viability within the printed structure was similarly high and the appearance of cells was indistinguishable to that of day 0 (Figure 6b,c). Whilst the percentage viability of cells was maintained at a level of 80% or greater throughout the 14-day culture period (Figure 6b), there was a clear decline in the number of viable cells after 4 days of culture, indicating that cells had died but could not be accounted for by EthD-1 staining (Figure 6c). The decline in viable cell number was most pronounced in 5% HV-alginate bioinks that displayed a considerable loss of viable cells in both non-stored and stored conditions at day 4. The number of viable cells in 4% HV-alginate bioink constructs were retained at day 4, with no considerable difference between stored and non- stored samples. By day 7, however, viable cell density had decreased in both samples but this was more pronounced in samples that had previously been stored.

### 4. Discussion

3D bioprinting as a tissue engineering tool is being used more prevalently in laboratories around the world due to the high levels of control over the spatial deposition of the cell-laden bioinks it provides. Although many types of bioprinting exist, extrusion based bioprinting is used the most, because of its relatively low cost, good post-print cell viability, a variety of readily available commercial printers and bioinks as well as the ability to combine several materials at once. The technique relies on extrusion of the material through a sub-millimetre orifice by air (pneumatic), piston or syringe-driven systems leading to layer-by-layer deposition of the extruded filaments onto a flat print bed and subsequent construct generation. To ensure good print fidelity of the cell-laden construct the extruded bioink needs to possess high viscosity sufficient for the structure to retain its shape during and after printing. The opposite is true for hydrogels used in traditional tissue engineering approaches, where compliant gels with low viscosity prior to crosslinking are chosen to ensure efficient gaseous and nutrient exchange crucial for the maintenance of long-term cell viability. This difference in viscosity makes hydrogels used in traditional tissue engineering approaches incompatible with the requirements of the bioprinting platforms and thus not directly translatable. Therefore, complex rheological requirements need to be considered to ensure shape fidelity as well as adequate maintenance of cell viability. Examples of such bioink requirements include: suitable viscosity (moderate polymer concentration) to ensure even cell distribution within the ink and prevent cell sedimentation prior to printing; use of highly hydrated biomaterials to promote nutrient diffusion and high cytocompatibility; viscoelasticity of the materials to protect the encapsulated cells from shear stress as well as shear thinning properties to allow for mixing of cells with the biomaterial and ensure the material can be deposited through a small diameter orifice; yield stress to ensure precise and accurate extrusion of the cell-laden material, lack of thixotropic (reversible, time-dependent decrease in viscosity as a result of a fixed shear rate) properties; rapid gelation to ensure post-print construct shape fidelity and stability as well as the ability of the extruded material to rapidly adopt solid-like properties upon deposition onto the print bed prior to crosslinking to prevent filament merging that could lead to a reduction in the material porosity and subsequent detrimental outcomes on cell survival [15,21,22].

Shear-thinning behaviour of polymer solutions has previously been shown to be highly concentration dependent. Molecular weight distribution also affects shear-thinning properties of bioinks [23]. This is because in dilute solutions with low concentrations (for instance 1% alginate) little interaction between the polymer chains exists. In more concentrated solutions a greater degree of intermolecular chain interaction between the polymer chains takes place in the form of non-covalent bonding and chain entanglement. Length of the polymeric chains and plasticity of the polymeric backbone determine the degree of entanglement, which subsequently increases the viscosity of the resulting solution, as we have observed in the performed viscometry experiments. Alginate bioinks with higher concentrations exhibited greater viscosity. Increasing polymer concentrations causes a faster decrease in viscosity with respect to shear rate, which is a behaviour we have observed in this study. In terms of physiochemical parameters of the de-gelled and reconstituted with HV-alginate bioinks, the viscosity of the solutions increased with increasing concentration of HV-alginate added, as expected. Viscosity has been shown to decrease with increasing shear rate in the tested materials. This behaviour is normal since alginate has been previously shown to exhibit shear thinning properties and thus has been widely used as a bioink of choice in the field [24,25,26,27,28,29,30].

Rheological analysis of the materials provides quantitative information that will be valuable for determining the extrusion forces required to deposit a certain bioink. Characterization via frequency sweeps was performed to determine the behaviour of the bioinks. Less concentrated solutions (1, 1.5, 2% alginate) exhibited primarily liquid-like behaviour, evident by the loss modulus exceeding the storage modulus at all frequencies tested, alginate inks (2.5–3.5%) showed viscoelastic liquid behaviour with phase change points (where G″ > G′) occurring at lower frequencies as the concentration of the alginate bioink increased. Alginate solutions with higher concentrations (4–5%) exhibited solid-like gel behaviour as suggested by the storage moduli dominating over the loss moduli. Obtained loss and storage modulus values appear to be frequency dependent, meaning that both moduli increase with increasing shear frequency. This is particularly pronounced in low concentration bioinks and less so in highly concentrated solutions, as evident by the rapid increases of G′ and G″ of less concentrated alginates tested and more gradual increases in the storage and loss moduli of alginate inks with higher concentrations. Viscoelastic liquids are known to be frequency dependent whereas materials that exhibit gel-like properties are less frequency dependent [22,31,32]. Typically, viscoelastic liquids are beneficial for cell viability, however do not exhibit any shape fidelity during the printing process, whilst gel-like bioinks do have better print resolution yet are detrimental for cell survival. This is the case we have observed during the performed experiments, we found that a concentration of between 4 and 5% (*w*/*v*) HV-alginate facilitated both rapid mixing and the ability to bioprint structures with high shape fidelity. These were then crosslinked for a second time forming stable hydrogel constructs. In addition to this, hASCs that had previously been stored for 1 week at 15 °C could be printed using this process with high post-print viability. This demonstrated that the process of de-gelling, reformulation/mixing, extrusion, and re-gelling did not affect cell vitality through generating excessive shear stress, a major concern when formulating hydrogels for extrusion-based bioprinting [8]. However, it is possible that the reason for the observed decrease of viable cells after 4 days of storage (particularly in the 5% alginate condition) was due to the smaller pore size present at higher concentrations, restricting the mass transfer of waste and nutrients [33].

Despite the diversity of industrial application of alginate, generally hydrogels have weak mechanical properties, low electrical and thermal conductivity as well as no inherent antibacterial activity and no cell adhesion domains to facilitate cell-matrix interaction within the bio-printed constructs. Strategies to enhance physiochemical properties (e.g., mechanical, thermal, electrical as well as water sorption and diffusion) biological properties (e.g., biodegradation, antimicrobial activity, cell adhesion, proliferation and differentiation and immunoengineering strategies) as well as tuning the porosity of alginates have been comprehensively described by Hurtado et al., 2022 [34]. The authors refer to several studies that managed to achieve modifications of alginate solutions to achieve desired improvements and characteristics of the polymeric materials. For instance, improvements in mechanical properties of alginates have been achieved via the addition of other polymers, fibers and nanofibers, carbon nanomaterials and nanoparticles to the alginate stock solutions. Frequently described materials include chitosan, cellulose, cellulose nanofibrils, nanocrystalline cellulose, carboxyethyl cellulose, hydroxyethyl cellulose, hyaluronic acid, Graphene oxide nanofibers and many others [35,36,37,38,39,40,41,42,43]. In terms of augmentations of alginate with the aim to improve biological properties a variety of bioactive components (e.g., gelatin, catechol, polycaprolactone, RGD motifs and other peptides) have been added to the alginate stock solution or chemically conjugated to the alginate backbone to provide anchorage motifs with the aim to improve cell-matrix interaction [44,45,46,47]. Enhancements in alginate biodegradation have also been shown via the use of oxidizing agents such as sodium periodate. Oxidized alginates retain their capacity for gel formation, however by breaking the bonds between the carbons of the cis-diol group conversion of the confirmation of the molecule produces an open chain, which improves the biodegradation abilities of the alginate. For tissue engineering applications production of an alginate matrix with increased porosity to promote greater mimicry of the in vivo extracellular matrix simulating nutrient and gaseous exchange is required. Alginate based scaffolds with increased porosity with incorporation of poly(lactic-co-glycolic) acid, hydroxyapatite as well as poly(ethelene) oxide and pluronic F127 have also been successfully employed for tissue engineering applications to improve the biocompatibility of otherwise bioinert alginate matrices [44,48,49].

Although the biocompatibility of alginate is excellent, it is also bio-inert. Therefore, it does not support the long-term culture or proliferation of anchorage-dependent cells, such as mesenchymal stem cells [33]. Without cell–cell or cell-matrix interactions, cell death will occur via anoikis [50]. To overcome the issues arising from alginate’s bioinert properties, and to promote cell-matrix interactions, alginate must be modified with bioactive peptides or be mixed with other matrices. Increased cell viability and growth in bioinert hydrogels has been demonstrated with a number of modifications including covalent linkage of the RGD peptide motif [38,39,40] and incorporation of bioactive matrices including fibronectin [50], gelatin [33,51], and collagen [51,52]. In addition to this, alginate porosity can be increased with the incorporation of cellulose without affecting viscosity [44], or cell–cell/cell-matrix interaction can be promoted through the controlled degradation of the alginate scaffold through the simple supplementation of culture medium with sodium citrate [51].

Alginate offers the perfect base for easy modification to affect cell viability, growth and function, whilst retaining the physiochemical properties appropriate for printing. This offers considerable flexibility in our system where the viscosity enhancing alginate (HV-alginate) can be modified to direct cell behaviour post-printing, opening up the possibilities of producing an array of bioinks depending on the encapsulated cell types or desired biological functions.

### 5. Conclusions

For the future success of the clinical translation of 3D biofabrication, it is important to consider the logistics of where cells are going to come from, how they are going to be stored and transported, and how they are going to be conveniently processed before use. Here, we describe a straightforward, flexible process that combines the capacity for alginate to preserve cell viability and functionality during storage, with its suitability for use as a bioink. Upon addition of the HV alginate to the dissolved stored samples the optimal results in terms of printability were obtained with 4–5% alginate bioinks (thread width 0.3–0.5 mm). Cell viability (assessed by CAM/EthD-1, live/dead staining) of 88 ± 18%, *p* < 0.05, was achieved in 4% and 5% alginate bioinks during the 2-week culture period. No significant differences in cell viability were observed between the samples stored at CRT and the non-stored control (*p* < 0.05), suggesting high suitability of the developed system for on-demand bioprinting and long-term culture of 3D bioprinted cell-laden constructs. When brought together, storable cell-laden bioinks could rapidly accelerate the translation of bioprinting in biomedicine alongside ongoing technological advances.

## Figures and Tables

**Figure 1 bioengineering-10-00023-f001:**
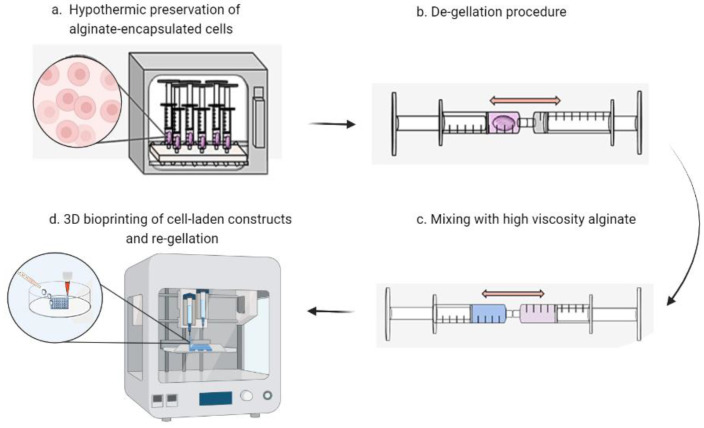
**Schematic illustrating the process for printing stored cells.** (**a**) hASCs encapsulated in BeadReady™ are stored for 1 week at controlled room temperature. (**b**) Alginate beads containing cells are dissolved using sodium citrate. (**c**) High viscosity alginate (HV-alginate) is mixed with the dissolved gel to increase ink viscosity. (**d**) Cell-laden bioink is extruded using a bioprinter and re-gelled using calcium chloride. Figure created with BioRender.com.

**Figure 2 bioengineering-10-00023-f002:**
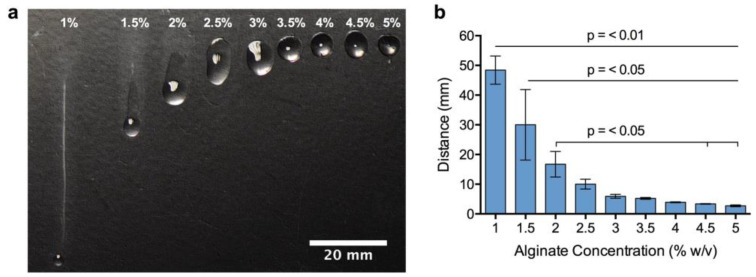
**Comparative viscosity of different bioink formulations.** (**a**) Calcium alginate beads were dissolved in 50 mM sodium citrate before mixing with HV-alginate to reach final concentrations of between 1 and 5%. 300 µL drops were placed on a glass slide for 15 min and relative viscosity was assessed by the distance the drop fell (**b**). Bioinks able to hold their shape were considered suitable for printing. Values are presented as means ± SD from 3 independent experiments. *p* < 0.05 was considered significant.

**Figure 3 bioengineering-10-00023-f003:**
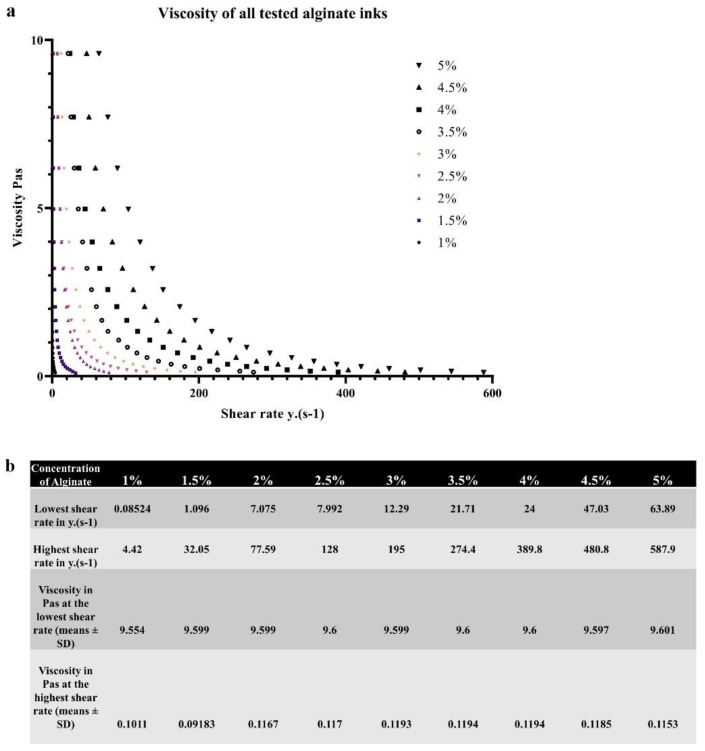
**Assessment of viscosity of different bioink formulations.** (**a**) Illustrates the dynamic viscosity of different bioink formulations across a range of shear rates. Figure legend corresponds to percentages of different alginates tested. (**b**) Displays the data from (**a**) in a table for easier comparison.

**Figure 4 bioengineering-10-00023-f004:**
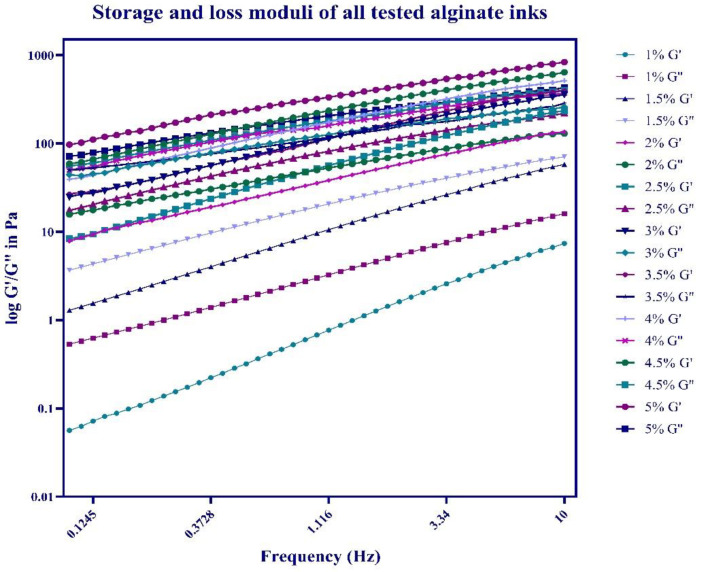
**Assessment of rheological parameters of different bioink formulations.** The storage (G′) and loss (G″) moduli (in Pa) obtained during oscillation frequency sweep of all the bioinks tested are presented. Figure legend corresponds to percentages of different alginates tested. Results were collected from six different samples and are presented as means ± SD. Error bars are not visible on the graph due to small degrees of variation.

**Figure 5 bioengineering-10-00023-f005:**
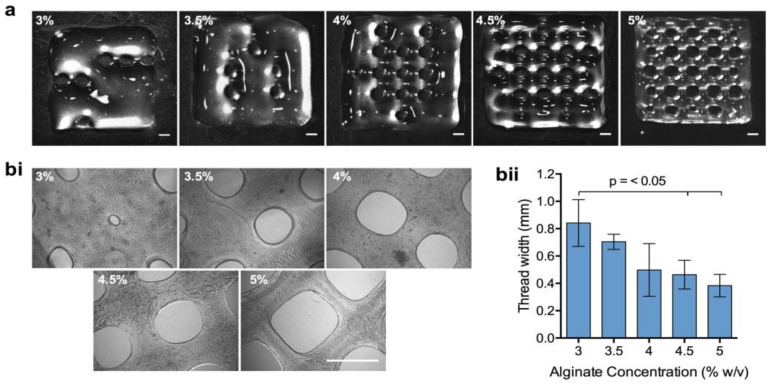
**Printability of different bioink formulations.** Calcium alginate beads were dissolved in 50 mM sodium citrate before mixing with high viscosity alginate (HV-alginate) to reach final concentrations of between 3 and 5%. Lattice constructs were printed using an extrusion-based bioprinter (**a**) before gelation and examination of print fidelity by phase-contrast microscopy (**bi**). Print fidelity was quantified by measuring thread width (**bii**). Values are presented as means ± SD from 3 independent experiments. *p* < 0.05 was considered significant. Scale bars = 2 mm.

**Figure 6 bioengineering-10-00023-f006:**
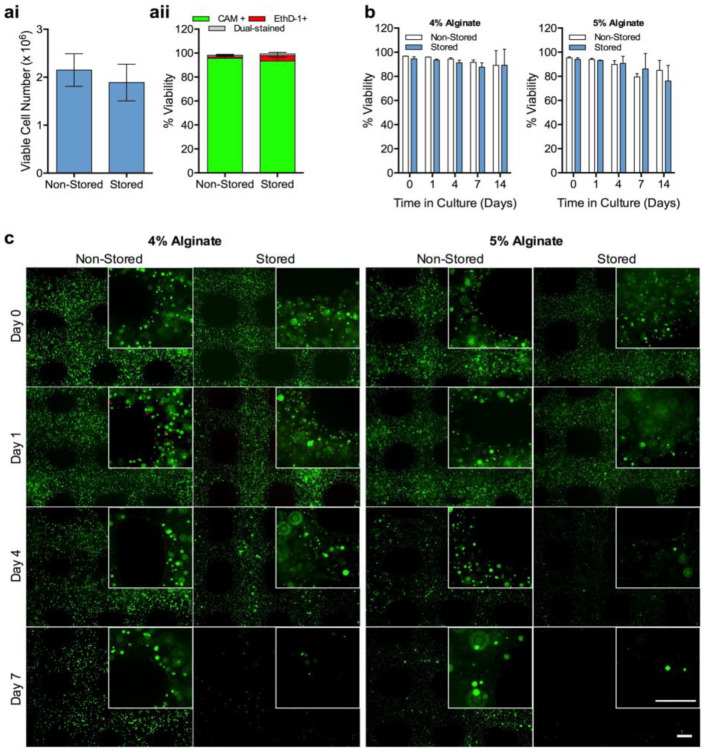
**3D bioprinting of storable cell-laden bioinks**. 2 × 10^6^ hASCs were encapsulated in calcium alginate beads and either used directly or stored for 7 days at 15 °C. Gels were dissolved in 50 mM sodium citrate before assessing viable cell recovery (**ai**) and % cell viability (*p* < 0.05) (**aii**) from dissolved gels (*p* < 0.05). Dissolved gels were mixed with HV-alginate, to reach final concentrations of either 4 or 5%, before printing lattice constructs using an extrusion-based bioprinter. Constructs were maintained under normal culture conditions for up to 14 days and percentage cell viability (% viability) (**b**) was calculated from live/dead (CAM/EthD-1) staining (**c**). Values are presented as means ± SD from 3 separate donors. *p* < 0.05 was considered significant. Scale bars = 0.5 mm.

**Table 1 bioengineering-10-00023-t001:** **Rheological parameters of different bioink formulations.** The storage (G′) and loss (G″) moduli (in Pa) as well as phase angle values at the lowest (0.1 Hz) and highest (10 Hz) shear frequencies are presented.

Concentration of Alginate	1%	1.50%	2%	2.50%	3%	3.50%	4%	4.50%	5%
Storage modulus (G’) in Pa at 0.1 Hz (means ± SD)	0.056	1.289	7.856	8.325	24.802	26.802	39.217	59.03	96.8
Storage modulus (G’) in Pa at 10 Hz (means ± SD)	7.362	57.860	135.100	234.700	357.500	385.375	512.837	640.3	835.1
Loss modulus (G") in Pa 0.1 Hz (means ± SD)	0.532	3.670	15.700	17.400	44.352	49.770	50.077	56.180	71.74
Loss modulus (G") in Pa at 10 Hz (means ± SD)	15.980	70.630	129.500	219.100	258.965	284.550	371.4	402.75	432.5
Phase angle (δ) in degrees at 0.1 Hz (means ± SD)	83.980	78.980	73.980	70.640	65.7	63.58	53.7	50.55	45.39
Phase angle (δ) in degrees at 10 Hz (means ± SD)	65.270	60.270	55.270	50.670	43.04	38.89	34.87	30.99	25.99

## Data Availability

Data available on request due to restrictions. The data presented in this study are available on request from the corresponding author. The data are not publicly available.

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
