# Peer review of "Storable Cell-Laden Alginate Based Bioinks for 3D Biofabrication"

_bioengineering, 2022, doi:10.3390/bioengineering10010023_

Round 1
Reviewer 1 Report
The manuscript entitled “Storable cell-laden alginate based bioinks for 3D biofabrication” provides a very interesting study for the scientific community. However, several points must be addressed before being suitable for publication:
- The abstract should highlight the novelty of this work
- The fact that alginate biopolymer can be enhanced by following several strategies developed so far should be mentioned in the introduction section or discussion section ( https://doi.org/10.3390/ijms23094486)
- The conclusion should be improved providing also quantitative results
Reviewer 2 Report
Presented article “Storable cell-laden alginate based bioinks for 3D biofabrication” shows the results of the application of sodium alginate with various concentration to encapsulate human adipose-derived stem cells for 3D biofabrication. Despite of the fact that the subject is interesting, the presentation of obtained results is very poor. The authors limit themselves mainly to the description of the obtained results rather than to their analysis. There is a lack of scientific aspect and explanation why certain phenomena occur during the process. I recommend to accept the manuscript after major revision.
The remarks and comments are the following:
3.1 Effect of alginate concentration on bioink viscosity
Some of the information in paragraph 3.1 overlaps with the information in paragraph 2. Please remove this information and focus on the measurement results. In addition, please put the measurement results in a table. Listing all the achieved results in the text is unreadable and does not show the trend between the used alginates with different concentrations.
3.3 3D bioprinting of storable cell-laden bioinks
Lines 347-354 - repeated sample preparation procedure. Please remove this section because the same information is included in paragraph 2.4.
Please also match the font to the rest of the text. Paragraph 3.3 was written in a smaller font.
4. Discussion
The considered paragraph has no nature of discussion in general. Information from lines 380 to 412 should be moved to Introduction. Lines 412 to 447 contain information about the obtained results, which should go in the Results paragraph. The discussion should contain scientific elements. For example: why does the concentration of alginate affect the values of the obtained parameters? What concentration of alginate is optimal for living cells? and so on.
Conclusion
Overall, Conclusion is a further discussion of the Abstract. It should contain all the information contained in the Abstract, together with its expansion and the values of the most relevant parameters. Please rewrite Conclusion.

Reviewer 3 Report
The authors have successfully demonstrated the use of alginate as a printable bioink and a cytopreservable cell scaffold for biofabricaiton applications. The experimental designs were well constructed and explained. The writing and narrative style were smooth and attractive to readers. This manuscript well served as a follow-up study of the authors' pervious investigation of the alginate's hypothermic preservation functions as reference 11 described. It was an overall well-prepared manuscript with some minor concerns listed below.
Figure 3 displayed some rheology characterization of alginate bioinks, however, I would consider the discussion remained week as only the shear-thining property was demonstrated. I would recommend this review paper regards to bioink rheology. (APL Bioeng. 5, 011502 (2021); https://doi.org/10.1063/5.0031475) The oscillaroty stress experiment could be added. Also, the results of frequency sweep should be displayed in one figure rather than two separated G' and G'' figures. In this way, the phase change points could be clearly shown in one graph, and the phase angle graph may not be too necessary.
Figure 3 needed to be remade by scientific graphing tools such as Origin or Prism rather than Excel. Log scales should be used. Legends should be remade. Repetative samples should be included for each concentrations, error bars should be added.
Figure 5 ai & aii, the p value between groups should be added.
Line 349, a few words could be added to describe in what conditions is non-stored.
Round 2
Reviewer 1 Report
The authors have addressed well all the reviewer's comments and the manuscript is now ready for publication
Reviewer 2 Report
Presented revised article “Storable cell-laden alginate based bioinks for 3D biofabrication” shows the results of the application of sodium alginate with various concentration to encapsulate human adipose-derived stem cells for 3D biofabrication. The authors took into account the reviewer's comments and added the relevant text. I suggest to accept the article in the present form.